# The Incidence and Severity of Myopia in the Population of Medical Students and Its Dependence on Various Demographic Factors and Vision Hygiene Habits

**DOI:** 10.3390/ijerph20064699

**Published:** 2023-03-07

**Authors:** Paweł Oszczędłowski, Przemysław Raczkiewicz, Piotr Więsyk, Kinga Brzuszkiewicz, Maria Rapa, Anna Matysik-Woźniak, Grzegorz Zieliński, Maksymilian Onyszkiewicz, Krzysztof Marek Rękas, Inga Makosz, Małgorzata Latalska, Aleksandra Czarnek-Chudzik, Jan Korulczyk, Robert Rejdak

**Affiliations:** 1Students’ Scientific Association at the Chair and Department of General and Pediatric Ophthalmology, Medical University of Lublin, 20-079 Lublin, Poland; 2Chair and Department of General and Pediatric Ophthalmology, Medical University of Lublin, 20-079 Lublin, Poland; 3Department of Sports Medicine, Medical University of Lublin, 20-093 Lublin, Poland; 4Department of Didactics and Medical Simulation, Medical University of Lublin, 20-093 Lublin, Poland

**Keywords:** myopia, refractive error, autorefractometry, COVID-19, visual work, medical students, online learning

## Abstract

(1) Background: Myopia is one of the leading causes of visual impairment. Visual work and usage of electronic devices are known risk factors of myopia. Many education systems were forced to apply online and hybrid teaching methods, to reduce the number of new cases of COVID-19. Medical students are a population well-known for intense visual work in the form of learning; (2) Methods: Visual acuity and refractive error were measured in the population of medical students. Participants also filled out the survey that included their population characteristic and their habits related to the hygiene of vision; (3) Results: We found a correlation between the age of the first diagnosis of myopia and current values of refractive error. The majority of participants believe that the COVID-19 pandemic had an impact on the health of their vision. Among methods of studying, usage of the computer screen was less preferred by myopic students; (4) Conclusions: In the population of medical school students in Eastern Poland, visual acuity was lower than 1.0 in 232 (52.97%) in the right eye and 234 (53.42%) in the left eye. Early recognition of refractive error has influenced its current values. Among methods of studying, usage of the computer screen was less preferred by myopic students. More population-based studies should focus on the impact of the COVID-19 pandemic on the health of vision.

## 1. Introduction

Uncorrected refractive error is the major cause of visual impairment (53%) and the second leading cause of blindness worldwide [1,2]. About a quarter of the world’s literate and educated population have a refractive error [2]. In addition, refractive error is responsible for most cases of blindness or reduced visual acuity in children in many countries. It has been found that uncorrected refractive error has extensive social and economic effects, such as reducing the educational and employment opportunities of the active population, healthy people and the community, social isolation, increased morbidity, and economic distress [3]. Medical professions, in particular, require good health and proper vision.

A refraction error is the inability of the eye to form a properly focused image on the macula located in the central part of the retina without accommodating [4].

Approximately 1.6 billion people in the world are affected by myopia [2,5]. Over the past 50 years, the frequency of myopia has already grown twice in Europe and the United States [6]. This fact makes it necessary to broaden the analysis of pathophysiology and look for ways to limit the progression of this error.

Myopia is a state in which the image focuses in front of the retina of the non-accommodating eye [7]. Despite effective correction methods, myopia remains a serious medical problem. For example, in subjects with high myopia (>−6 Diopters (D), accounting for 10% of all myopia cases, the risk of developing glaucoma rises 14 times, for cataracts 3.3 times, and in myopia above −8 D, the risk of retinal detachment increases even 7.8 times [8].

Excessive elongation of the eyeball is the most important pathological mechanism in myopia. The state in which the eyeball’s length is increased is called axial myopia. The elongation is often accompanied by bioelectrical activity [9] and the thickening of cranial and cervical spine muscles [10]. The multifactorial etiology of myopia depends on genetic and lifestyle factors. Among lifestyle factors, it is worth noting that myopia is more prevalent in children from developed countries. This phenomenon could be explained by the fact that they spend more time on indoor activities that can be put under the term “near work”: usage of a computer screen and learning, compared to their peers from less developed countries [11]. Another group exposed to increased amounts of near-visual work is university students [12]. The COVID-19 pandemic has resulted in increased amounts of screen usage in all age groups [13]. Online learning, increased digital device usage, and decreased physical activity during the COVID-19 pandemic were found to increase the risk of myopia progression [13,14,15].

Myopia onset occurs typically during childhood, teenage years, or adolescence. The strongest progression of eye growth is observed in early childhood [16]. Myopia becomes stable in almost 90% of cases at the age of 21, and at the age of 24, it can be applied to almost all myopes [16,17]. However, in a recent editorial, Iribarren et al. have pointed out that while the majority of research about myopia is targeted at school-age children, the usual group for myopia development, rapid progression of myopia is also possible in the third decade of life [18]. Moreover, the progression of myopia was already observed in university students [12]. As our study population consists of people aged 19–26, this paper helps to fill an important research gap.

The aim of the study is to assess the incidence and severity of myopia in the population of medical students and its dependence on various demographic factors and vision hygiene habits.

## 2. Materials and Methods

The study was conducted on the Caucasian population aged 19–26 from fields of study such as Medicine, Dentistry, Pharmacy, Physiotherapy, Medical Analytics, Dietetics, Biomedicine, Electro radiology, Dental hygiene, Cosmetology, Nursing, Midwifery, and Emergency medical service. All participants consented to the processing of personal data. The Bioethics Committee of the Medical University of Lublin has approved the protocol of the presented research (KE-0254/89/2021 date of approval 29 April 2021). Participants were informed about the study’s aim during the recruitment procedure, and written consent was obtained from all subjects involved. The inclusion criterion for participants was active student status at the Medical University of Lublin. The testing procedure included filling out a survey about habits related to the health of vision, visual acuity examination, and examination of the refractive defect. Vision acuity was tested using a Snellen chart over the exact distance of 5 m [19]. Refractive error was measured using an SVS-100S photoscreener (Spot Vision Screener) by the Welch Allyn, described in the literature as an acceptable method of screening refractive errors [20,21,22,23]. Participants were sitting upright, each in the same type of chair, while the person who was performing the procedure was standing at the distance recommended automatically by the auto refractometer. If the distance was too long or short, the photoscreener showed the notice to the examiner that there was a need to correct it. In each procedure, the autorefractometer was held at the recommended height of the participant’s eyes. The methodology was similar to the study performed by Maruyama et al. [24]. The procedure was applied in an environment recommended by the producer- in a darkened room from a distance of 1 m. The testing of the refractive defect was conducted without pharmacological dilation of the pupil, as described by the producer of the device. 

Data were analyzed using Statistica 13 software by StatSoft. Nominal variables (Belief about COVID-19 impact on the health of vision “Yes,” “No;” Belief about worsening of vision “Yes,” ”No,” Methods of studying “Textbooks and printed study materials,” “ Smartphones and tablets,” “Computer screen”) were analyzed using Pearson’s χ^2^ test. Ordinal variables (Place of residence in childhood “Countryside;” “City < 100,000 citizens”, “City > 100,000 citizens”; Age of diagnosis “<10 years old”, “>10 years old” were analyzed with non-parametric U-Mann Whitney and Kruskal-Wallis tests appropriate for the number of groups. Numerical variables had their normality of distribution verified with Shapiro-Wilk test in all compared groups. Refractive error values were also analyzed with U-Mann Whitney and Kruskal-Wallis tests. The threshold of statistical significance was set at *p* < 0.05. 

## 3. Results

### 3.1. General Characteristics

Among 438 participants, 333 (~76%) were women, 92 (~21%) were men, and 13 (~3%) participants preferred not to specify their gender. Visual acuity was lower than 1.0 in 232 (52.97%) in the right eye and 234 (53.42%) in the left eye. Binocular myopia (refractive error lower than −0.5 D in both eyes) was present in 119 (27.17%) participants. Monocular myopia was present in 147 (33.56%) right eyes and 142 (32.42%) left eyes. The biggest group of participants declared living in the countryside in their childhood (40.41%; 177), while 35.39% (*n* = 155) had lived in cities below 100,000 citizens and 24.20% (*n* = 106) in cities above 100,000 inhabitants. As mentioned previously, participants have filled out the survey, the results of which are displayed in Table 1. 

### 3.2. Methods of Learning

In non-myopic participants, usage of computer screens for learning is almost evenly split: 50.47% (*n* = 161) participants use them, while 49.53% (*n* = 158) declared no usage. In the myopic group, only 37.82% (*n* = 45) declared using computer screens for learning, and 62.18% (*n* = 74) did not use them. The differences in computer screen usage between myopic and non-myopic participants have reached statistical significance (*p* = 0.03 in Pearson’s χ^2^ test). The usage of smartphones and tablets was similar for both groups (31.97% declared usage in the non-myopic group and 33.61% in the myopic group). The comparison did not reach statistical significance (Pearson’s χ^2^ test; *p* = 0.74). Textbooks and other printed learning materials were used by 33.86% of non-myopic participants and 42.02% of myopic participants. Differences in the usage of printed materials were also insignificant (Pearson’s χ^2^ test; *p* = 0.11). All differences in preferred methods of learning are displayed in Figure 1.

### 3.3. Characteristics of Myopic Population

The percentage distribution of places of residence in childhood was very similar for non-myopic and myopic participants (Countryside 40.41% vs. 41.18%, City below 100,000 citizens 35.39% vs. 36.13%, City above 100,000 citizens 24.20% vs. 22.69%). Those comparisons did not reach statistical significance in U-Mann Whitney test (*p* = 0.74). Among individuals with binocular myopia, 33.61% (*n* = 40) were diagnosed before the age of 10, while 66.39% (*n* = 79) were diagnosed after that age. Survey answers for the myopic group are shown in Table 2. 

### 3.4. Impact of COVID-19 Pandemic on Health of Vision

In the general study population, 46.12% of participants suspected deterioration of their vision during the 2 years of the COVID-19 pandemic. In the myopic group, the number rises to 78.15%. In the general population, 55.02% of participants believe that the COVID-19 pandemic had an impact on the health of vision. Among myopic participants, it rises up to 63.87%, while in the non-myopic group, only half (51.72%) believe in that statement. Although these results were deemed insignificant in Pearson’s χ^2^ test (*p* = 0.20), a positive correlation was found between the suspected deterioration of participants’ vision and their belief about the COVID-19 pandemic’s impact on the health of vision. In the group that suspects that their vision has worsened, 76.73% (*n* = 155) believe that the pandemic has affected their vision health, while in a group that denies a deterioration of vision in the last 2 years, it has reached only 36.44% (*n* = 86). The correlation was statistically significant in Pearson’s χ^2^ test (*p* < 0.0001).

### 3.5. Correlation of Refractive Error with Other Factors

Among the myopic participants (*n* = 119) mean refractive error was −2.41 D in the right eye and −2.27 D in the left eye. The median refractive error was −2.00 D in the right eye and −1.75 D in the left eye. The standard deviation was σ = 1.589 in the right eye and σ = 1.547. 

#### 3.5.1. Place of Residence in Childhood

Refractive error values depending on the place of residence in childhood are displayed in Table 3. In the W Shapiro-Wilk test, the hypothesis of normality of distribution was rejected in both eyes for all three types of residence. For both eyes, no correlation was found between the place of residence in childhood and the value of current refractive error (Kruskal-Wallis test, *p* = 0.54 in the right eye, *p* = 0.80 in the left eye).

#### 3.5.2. Age of Diagnosis

Correlation between age of diagnosis and value of current refractive error was found with the U-Mann Whitney test (U = 918.00; Z = −3.72; *p* < 0.01 for the right eye, U = 925.00; Z = −3.68; *p* < 0.01 for the left eye). Refractive error values depending on participants’ age of diagnosis are shown in Table 4 and Figure 2. The hypothesis of normality of distribution was rejected in both eyes for both age ranges (W. Shapiro Wilk test). 

#### 3.5.3. Beliefs about COVID-19 Impact and Refractive Error

No correlation was found between “Belief on the impact of the pandemic on the health of vision” and the value of refractive error in myopic participants in both eyes (U-Mann Whitney test, *p* = 0.92 for the right eye and *p* = 0.99 for the left eye).

## 4. Discussion

It is well known that the prevalence and progression of myopia are dependent on many factors, such as age, the onset of myopia, country, and ethnicity [25]. 

The incidence of myopia (27.17%) was comparable to the pre-pandemic data in young adults. In the Raine study (1344 participants the age of 20, Australia), myopia had a 25.8% of prevalence [26]. In a meta-analysis that included 61,946 adults, the general incidence of myopia in Europe was 24.2%, although the incidence in young adults was higher (47.2% in the group aged 25–29 [27]). In the Polish population, the mean prevalence in adults was similar (24.1%) [28]. That said, the mean prevalence of myopia in our study was slightly higher but similar to the Polish and European mean in adults. 

Although myopia progression in medical students has been suggested for a long time [29], the amount of studies is severely limited. Moreover, the grand majority of this research was conducted before the COVID-19 pandemic [30,31,32,33,34,35,36,37,38,39,40,41]. In Argentina, the pandemic has had an impact on myopia progression in newly developed myopia cases [42]. Table 5 shows the prevalence of myopia in medical students in our study and other papers.

As shown in Table 5, our results were similar to the data from Turkey, Saudi Arabia, and Bangladesh. Data from other countries, especially China, shows a much higher prevalence. This topic should be addressed in further studies, focusing not only on the prevalence of myopia but also on the factors that influence it in the study population. As the data from the literature suggests that even 2 years of studies can be enough for myopia to progress [12,39], there is also a need for regular myopia screening in students.

Risk factors for high myopia at adult age are a young age of onset and a fast progression rate during childhood [5,6,7,16]. Our data support this statement- medical students that have been diagnosed with myopia before the age of 10 currently have higher myopia than ones diagnosed after that age (Table 4). Being exposed to the effects of the disease for a longer time results in an increased risk of high myopia [44,45]. In 2019, the American Academy of Ophthalmology created the Task Force on Myopia. One of its tasks is to encourage “the screening and early diagnosis of children with myopia to maximize the benefit of myopia control interventions” [46]. The results show that participants that were diagnosed before the age of 10 had higher myopia during the period of studies, reinforcing the statement about the necessity of early diagnosis. Many authors agree that the identification of early myopia cases is a matter of great importance, as it helps with the early application of the countermeasures, including the correction of vision defects [18,47,48]. Screening of myopia in children, reinforced with analysis of the family history of myopia in cases at risk of developing high myopia, is proposed as one of the solutions [18,47,48].

We have found that place of residence in early childhood has no impact on the frequency and magnitude of myopia in this population. Results of different studies show conflicting results regarding the association between myopia and place of residence in childhood [47,49,50,51]. In a study conducted on 338 freshman students in Saudi Arabia (162 males, 176 females), place of residence had no effect on myopia progress [49]. In the Chinese study on 2454 children, myopia was associated with spending early school years in urban environments (50%) more than with rural ones (33%) [47]. In the Malaysian study, 4634 children were assessed for the presence of refractive error. Myopia risk was lower in the rural regions than in urban regions both in China and Mexico [50,52]. The high prevalence of myopia among medical students (27.17%) requires further studies to explore more about the clinical characteristics and risk factors of myopia. In a study from Argentina, near extensive work during the school period of life was associated with myopia onset [53]. As medical students are usually active, engaged learners from young age, this may influence the prevalence of myopia in the whole group, despite their original place of residence. Further research should include more diverse groups, including people from other fields of study. Students with myopia use a computer screen to study less frequently than their non-myopic peers. The majority of medical students believe that the COVID-19 pandemic has had an impact on their vision, and almost half of them believe that their vision has worsened in the period of the pandemic. Our results show that more respondents learn with the use of a computer (46%) than with textbooks, and printed study materials (36%), which may be caused by increased digital learning during COVID-19 pandemic [13]. Forementioned study by Ma D et al. showed that in the surveyed population, during the 7 months of home quarantine caused by the pandemic, myopia developed on average by −0.9 D, while before COVID-19, myopia progression was on average −0.3 D yearly [14]. Myopic individuals were also more prone to declare that online and hybrid learning may have adverse effects on their health of vision. This brings forward a need for further studies on a larger population.

### Limitations of the Study

The main limitation of the study was the fact that the data about the earlier history of myopia in participants were based on their answers to the survey, not medical documentation. There is also no data about previous ophthalmological examinations of the same group before the pandemic. There is also a need for a longitudinal observational study focused on the influence of intense near-visual work on the prevalence and severity of myopia, for there is no data about the time period necessary for differences to occur. The study population consisted mainly of women, and the female gender is one of the well-established risk factors for myopia [54]. Another important fact was that the measurements of refractive error were non-cycloplegic, performed using a photoscreener, not an autorefractometer. Although, we would like to point out that the factor that has influenced our decision is that our study was planned at the height of the COVID-19 pandemic and carried out still during the pandemic Usage of a photoscreener that allows us to scan refractive errors from a longer distance than the autorefractometer was a safer option to screen the group consisting of hundreds of subjects that could be potentially infected. What is more, our study shows that performing a myopia screening is possible even during events such as the COVID-19 pandemic. Additionally, the high sensitivity and specificity of the used photoscreener were confirmed by former studies [20,21,22,23], making it a potential tool for the population’s general refractive error assessment. The study was not fully randomized. As the study was conducted only among medical college students, this may have influenced their beliefs about the impact of the COVID-19 pandemic on the health of vision. This may be due to their high level of awareness and knowledge of risk factors affecting the deterioration of the quality of vision. In order to increase the credibility of obtained results, we recommend focusing on research on another ethnic group and also on non-medical college students.

## 5. Conclusions

In the population of medical school students in Eastern Poland, visual acuity was lower than 1.0 in 232 participants (52.97%) in the right eye and in 234 participants (53.42%) in the left eye. Based on the data from the study, it cannot be concluded clearly that online and hybrid learning had an impact on the exacerbation of myopia. The prevalence of myopia was similar to the Polish and European mean prevalence of myopia in adults. What should be concerning is the fact that more than half of medical students have visual acuity lower than 1.0. Medical students are a group exposed to many risk factors of myopia and its progress. Higher values of refractive errors can affect their performance in the usage of medical equipment, altering the choice of their preferred medical specialties. It is one of the reasons why myopia should be considered a major social problem. There is a need for a larger study that includes refractive error examinations in regular periods of time during studies. Education about myopia and its risk factors is an urgent need. Early diagnosis and education are crucial for the health of vision in both children in adults. Myopia screening and education about this disease should be coordinated by the government, as myopia is an important social problem. 

## Figures and Tables

**Figure 1 ijerph-20-04699-f001:**
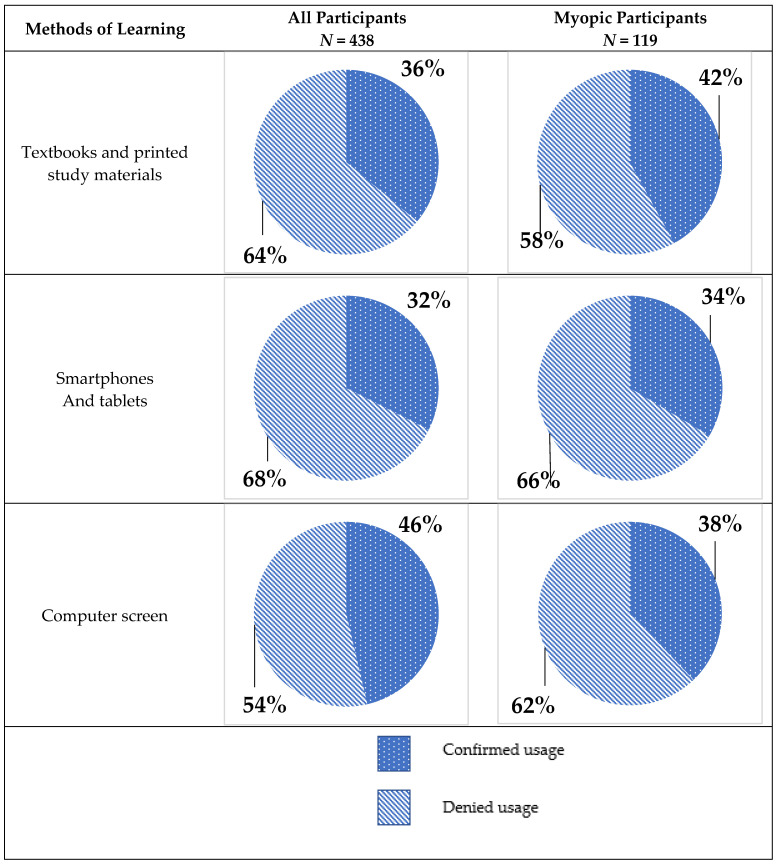
Usage of technical devices and printed study materials in the whole study population and myopic group.

**Figure 2 ijerph-20-04699-f002:**
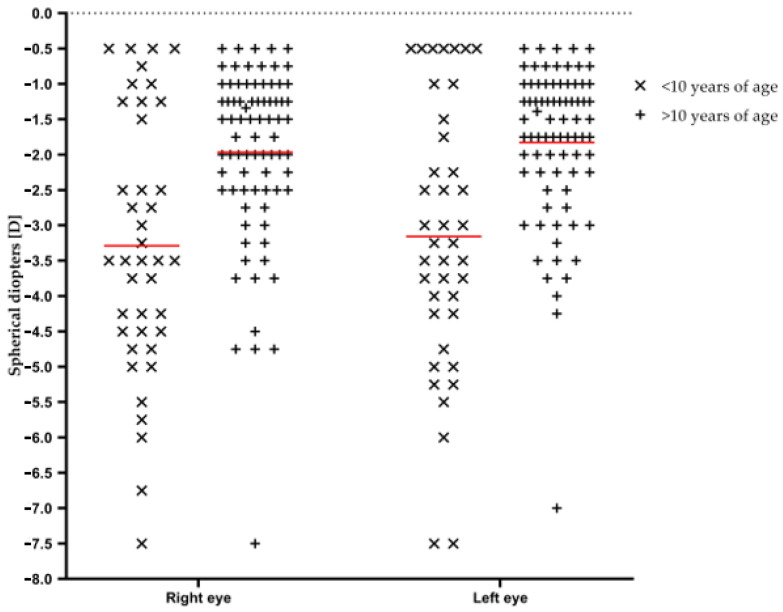
Refraction error in participants grouped by the age of diagnosis Right eye and Left eye.

**Table 1 ijerph-20-04699-t001:** Answers to survey questions are applicable to the whole population.

Survey Question	Answer	*N*	%
Where did you live in your childhood and early adulthood?	Countryside	177	40.41
	City < 100,000 citizens	155	35.39
	City > 100,000 citizens	106	24.20
How much time daily do you spent on a near vision work	<1 h	276	63.01
	1–2 h	5	1.14
	2–3 h	22	5.02
	3–4 h	47	10.73
	4–5 h	88	20.09
Do you believe that your eyesight has worsened in the last 2 years?	Yes	202	46.12
	No	236	53.88
Do you believe that COVID-19 pandemic had an impact on health of vision	Yes	241	55.02
	No	145	33.11
	Not answered	52	11.87

**Table 2 ijerph-20-04699-t002:** Survey answers in the myopic group.

Survey Question	Answer	*N*	%
Where did you live in your childhood and early adulthood?	Countryside	49	41.18
	City < 100,000 citizens	43	36.13
	City > 100,000 citizens	27	22.69
At which age were you diagnosed with myopia	<10 years of age	40	33.61
	>10 years of age	79	66.39
How much time daily do you spent on a near vision work	<1 h	76	63.87
	1–2 h	2	1.68
	2–3 h	6	5.04
	3–4 h	14	11.76
	4–5 h	21	17.65
Do you believe that your eyesight has worsened in the last 2 years?	Yes	93	78.15
	No	26	21.85
Do you believe that COVID-19 pandemic had an impact on health of vision	Yes	76	68.87
	No	34	28.57
	Not answered	9	7.56

**Table 3 ijerph-20-04699-t003:** Refractive error in myopic students depends on their place of residence in childhood and early adolescence.

Eye	Place of Residence	*N*	Mean	Median	Standard Deviation	Test H	*p*
Right	Countryside	49	−2.22 D	−2.00 D	1.50	1.22	0.54
	City < 100,000	43	−2.56 D	−2.00 D	1.71		
	City > 100,000	27	−2.53 D	−2.25 D	1.56		
Left	Countryside	49	−2.14 D	−1.75 D	1.42	0.39	0.82
	City < 100,000	43	−2.36 D	−1.75 D	1.70		
	City > 100,000	27	−2.38 D	−2.00 D	1.54		

**Table 4 ijerph-20-04699-t004:** Spherical refractive error in myopic students depending on their age of diagnosis.

Eye	Age of Diagnosis	N	Mean	Median	Standard Deviation	Test Z	*p*
Right	<10 years	40	−3.29 D	−3.5 D	1.84	−3.72	<0.01 (0.0002)
	>10 years	79	−1.97 D	−1.5 D	1.23		
Left	<10 years	40	−3.16 D	−3.25 D	1.90	−3.68	<0.01 (0.00022)
	>10 years	79	−1.83 D	−1.5 D	1.10		

**Table 5 ijerph-20-04699-t005:** Percentage of myopic medical students in different countries [30,31,32,33,34,35,36,37,38,39,40,43].

Study and Year	N of Participants	% of Myopic Medical Students	Country
Our study	438	27.17%	Poland
Abuallut et al., 2020 [34]	447	33.8%	Saudi Arabia
Duan et al., 2019 [40]	318	92.8%	China
Shi et al., 2018 [31]	3654	92.52%	China
Wang et al., 2017 [35]	11,138	70.50%	China
Grigoraş et al., 2016 [36]	576	73.8%	Romania
Lv et al., 2013 [39]	2053	84.1%	China
Akhanda et al., 2010 [43]	175	38.29%	Bangladesh
Jacobsen et al., 2008 [33]	151	43%	Denmark
Onal et al., 2007 [30]	207	32.9%	Turkey
Fledelius et al., 2000 [38]	294	50%	Denmark
Midelfart et al., 1992 [32]	140	50.3%	Norway
Chow et al., 1990 [37]	128	72%	Singapore

## Data Availability

Not applicable.

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
