# Peer review of "The Incidence and Severity of Myopia in the Population of Medical Students and Its Dependence on Various Demographic Factors and Vision Hygiene Habits"

_ijerph, 2023, doi:10.3390/ijerph20064699_

Round 1

Reviewer 1 Report

Complete dates from latinamerica, there are, a lot of people with error refractive. for discussion

Author Response

  1. ,, Complete dates from latinamerica, there are, a lot of people with error refractive. for discussion’’

We would like to kindly thank the Reviewer 1 for a helpful suggestion. The data from studies from Latin-American countries has been added to the lines: 261-262, 300-302, 306-307.

As suggested by the Reviewers, the article passed the reviews by a professional working with medical English and scientific English, employed in the Medical University of Lublin. We have provided the details to the Editor.

Please, see the attachment- Lines in which changes were applied refer to the version of the manuscript sent to the Reviewers in an attachment to the ,,Authors’ Reply to the Reviewer Report’’ (titled ,, ijerph-2216382 - version for the Reviewers’’). In the version of the paper sent to the reviewers the linguistic corrections are marked with blue colour. Changes in the text proposed by the reviewers are marked in red. The text without the colour markings is accessible as the Submitted Revised Manuscript, titled ,,ijerph-2216382’’.

Reviewer 2 Report

This is a study of the frequency of refractive error in Polish young adults in the medical professions. Much of this article is a review of the literature with very little original research. What is the original is the frequency of myopic refractive error and the exploration of some factors such as place of residence (city or countryside), age of incidence of myopia, and amount and type of near work. About a third of the study subjects were myopic, which was not surprising. Those who developed myopia earlier were more myopic, which was not surprising. There was no relationship between living in the city or countryside. Myopic students were less likely to use the computer screen for study. I would avoid the use of the terms "organ of vision" and "place of accommodation" because it could be confused with the focusing mechanism of accommodation. The manuscript must be reviewed by a native English speaker for corrections. The way I see it, this manuscript offers very little in the way of innovative research and I would not give it high priority for publication. Furthermore, I did not see evidence of informed consent and protection of human subjects, although I will grant that the risk of harm would have been very low.

Author Response

  1. ,,This is a study of the frequency of refractive error in Polish young adults in the medical professions. Much of this article is a review of the literature with very little original research. What is the original is the frequency of myopic refractive error and the exploration of some factors such as place of residence (city or countryside), age of incidence of myopia, and amount and type of near work. About a third of the study subjects were myopic, which was not surprising. Those who developed myopia earlier were more myopic, which was not surprising. There was no relationship between living in the city or countryside. Myopic students were less likely to use the computer screen for study.’’

We would like to kindly thank Reviewer 2 for helpful suggestions about improvements in our paper. Firstly, we would like to point out that although we agree with the fact that there are many epidemiological studies about frequency of myopia, our study helps to fill the important gap of knowledge, as studies of this type are rare in the population of medical students.

  1. ,, I would avoid the use of the terms "organ of vision" and "place of accommodation" because it could be confused with the focusing mechanism of accommodation. The manuscript must be reviewed by a native English speaker for corrections.’’

Thank you for your reviewer's feedback and suggestions. We have removed the phrases ,,organ of vision’’ and ,,place of accomodation’’ that could have been misleading for a potential reader

  1. ,,The way I see it, this manuscript offers very little in the way of innovative research and I would not give it high priority for publication.’’

As forementioned, we would like to point that in our work we have focused on a population that has not been screened before: medical students (young adults) in Eastern Europe.  Secondly, in our paper we have used a novel screening method for myopia in adults: a photoscreener. The device was previously mostly used for studies on paediatric population (10.3390/ijerph19148655.   10.1016/j.jaapos.2018.09.012       10.1016/j.jaapos.2014.07.176) To the best of our knowledge, our paper could bring attention to the health of vision of medical students and usefulness of Spot Vision Screeners for refractive error screening in adults.

  1. ,,Furthermore, I did not see evidence of informed consent and protection of human subjects, although I will grant that the risk of harm would have been very low.’’

In the statement section there is information about obtaining approval from the bioethics committee (Lines 367-370).  We have also expanded the information about the Bioethics Committee approval and participants’ informed consent (Lines 98-102).

As suggested by the Reviewers, the article passed the reviews by a professional working with medical English and scientific English, employed in the Medical University of Lublin. We have provided the details to the Editor.

Please, see the attachment- Lines in which changes were applied refer to the version of the manuscript sent to the reviewers in an attachment to the ,,Authors’ Reply to the Reviewer Report’’ (titled ,, ijerph-2216382 - version for the Reviewers’’). In the version of the paper sent to the reviewers the linguistic corrections are marked with blue colour. Changes in the text proposed by the reviewers are marked in red. The text without the colour markings is accessible as the Submitted Revised Manuscript, titled ,,ijerph-2216382’’.

Reviewer 3 Report

This study has a correct design and analysis. However some details could be improvement:

Myopia is not the second leading cause of blindness worldwide, is the glaucoma, because blinsness and low visión implies a reduced value of visual acuity with the best prescription.

In the introduction the information about near vision task is repeated in different paragraph. Besides, introduction is confusing in the age until myopia progress.

Line 90-94. Maybe, this information could be more useful in the discussion to compare with the study results.

Line 103, the number of participants are results, not methods, as the information in line 106-107.

Line 114, Spot Vision Screene (Welch Allyn) is not a autorefractometer is a photoscreener, the procedure to determine the refractive error is different and their precisión too. Besides, photorefraction is not the better method to determine the refractive error, is a method to perform visual screening because this measurement tend to be more negative than other methods.

Line 147, table 1 leyend should be in the same page to the table.

Figure 1 is blurred.

Line 180, only, may not be the right Word for the 46%.

Line 186, I believe that this p-value should only have hundredths.

Line 217-221, This information are results, not discussion.

Line 305, this information is reported in the 299.

The manuscript have some erratums:

Line 55, there is a space left.
Line 68, point missing.
Line 73, point missing.
Line 93, there is two spaces left.
Line 121, 241, incorrect use of script
Line 140, 171,180, 186, too many spaces
Line 246, incorrect use of “
Line 259, missing dot.
Line 276, our study?

Author Response

  1. ,,This study has a correct design and analysis. However some details could be improvement: Myopia is not the second leading cause of blindness worldwide, is the glaucoma, because blinsness and low visión implies a reduced value of visual acuity with the best prescription.’’

We would like to thank Reviewer 3 for helpful comments and suggestions for improvements in our paper. According to the paper ,, Global causes of blindness and distance vision impairment 1990–2020: a systematic review and meta-analysis’’ (DOI: 10.1016/S2214-109X(17)30393-5 ) refractive errors (2nd cause) preceded glaucoma (3rd cause) as a cause of blindness worldwide (Lines 35-36).

  1. In the introduction the information about near vision task is repeated in different paragraph. Besides, introduction is confusing in the age until myopia progress.

We agree with the reviewer. The paragraphs related to near visual work were merged into one, in the lines 63-72. Alike, the paragraphs concerning the matter of myopia progress age were merged into a cohesive paragraph describing the matter 80-91.

  1. Line 90-94. Maybe, this information could be more useful in the discussion to compare with the study results.

We agree with the reviewer. This has been reassigned to the discussion. (Moved to the lines 309-316).

  1. Line 103, the number of participants are results, not methods, as the information in line 106-107.

We agree with the reviewer. This has been reassigned to the results.

  1. Line 114, Spot Vision Screene (Welch Allyn) is not a autorefractometer is a photoscreener, the procedure to determine the refractive error is different and their precisión too. Besides, photorefraction is not the better method to determine the refractive error, is a method to perform visual screening because this measurement tend to be more negative than other methods.

Thank you for your attention. The sentence has been corrected (Lines 108 and 112). The photoscreener that was used has a satisfactory detection rate.  

‘’The specificity of the Spot Vision Screener to detect refractive errors was found to be relatively high (>90%).’’ - 10.1016/j.jaapos.2018.09.012

It is also acceptable as an acceptable method of detecting a refractive defect-  10.3928/01913913-20200331-02  
We have added the information about previous uses of mentioned photoscreener in the lines 108-109.

  1. Line 147, table 1 leyend should be in the same page to the table.

We agree with the reviewer. It has been corrected.

  1. Figure 1 is blurred.

We agree with the reviewer. It has been corrected.

  1. Line 180, only, may not be the right Word for the 46%.

We agree with the reviewer. It has been corrected.

  1. Line 186, I believe that this p-value should only have hundredths.

We agree with the reviewer. P-values were corrected to p<0,01 in lines mentioned by the reviewer (232, 233, Table 4).

  1. Line 217-221, This information are results, not discussion.

We agree with the reviewer. The information was removed from the discussion section.

  1. Line 305, this information is reported in the 299.

We agree with the reviewer. It has been corrected

As suggested by the Reviewers, the article passed the reviews by a professional working with medical English and scientific English, employed in the Medical University of Lublin. We have provided the details to the Editor.

Please, see the attachment- Lines in which changes were applied refer to the version of the manuscript sent to the reviewers in an attachment to the ,,Authors’ Reply to the Reviewer Report’’(titled ,, ijerph-2216382 - version for the Reviewers’’). In the version of the paper sent to the reviewers the linguistic corrections are marked with blue colour. Changes in the text proposed by the reviewers are marked in red. The text without the colour markings is accessible as the Submitted Revised Manuscript, titled ,,ijerph-2216382’’.

Round 2

Reviewer 2 Report

The changes made have improved the manuscript.

Line 98. "has informed" could be "showed a notice to"

Line 117 "man" should be "men"

Line 155. "suspects" should be "suspected"

The most interesting thing about the study is that location of residence, countryside versus city, had no effect on myopia. That tends to be contrary to other studies. Second, early myopia incidence was a predictor for higher myopia later. That is a very common finding in myopia research.

Please acknowledge that (1) the refractive error measurement was non-cycloplegic and (2) a photoscreener was used. The WA Spot photoscreener was not intended to measure and report on refractive error. It was intended to be a screener - pass or fail. This group is using the photoscreener as a refractive error measurement device. Also, the WA Spot photoscreener was mostly intended for pediatric use and your study sample is young adults.

Author Response

  1. The changes made have improved the manuscript.

We would like to thank Reviewer 2 for feedback about our revised paper.

  1. Line 98. "has informed" could be "showed a notice to"

Thank you for your attention. The sentence has been corrected

  1. Line 117 "man" should be "men"

Thank you for your attention. The sentence has been corrected

  1. Line 155. "suspects" should be "suspected"

Thank you for your attention. The sentence has been corrected

  1. The most interesting thing about the study is that location of residence, countryside versus city, had no effect on myopia. That tends to be contrary to other studies. Second, early myopia incidence was a predictor for higher myopia later. That is a very common finding in myopia research.

We would like to thank Reviewer 2 for feedback. As observed in a study in Argentina (10.1016/j.jaapos.2022.08.525), more visual work in form of the extracurricular learning activities in childhood is associated with increased prevalence of myopia. As medical students are usually active, engaged learners since young age, this may influence the prevalence of myopia in the whole group, despite their original place of residence. To explain this phenomenon in the manuscript, we have rewritten the paragraph about the place of residence in the discussion and provided additional information for potential readers [lines 235-251]. Additionally, according to the document about demographic trends in Poland, after the fall of socialism in Poland there was an increased population movement to the suburbs that are usually considered as ,,countryside’’ near the cities (10.4467/20833113PG.12.017.0658), which may influence the results. Despite the forementioned facts, early myopia was found to be the predictor for higher myopia- and as it supports the data from earlier studies, it is an important finding in this specific population.

  1. Please acknowledge that (1) the refractive error measurement was non-cycloplegic and (2) a photoscreener was used. The WA Spot photoscreener was not intended to measure and report on refractive error. It was intended to be a screener - pass or fail. This group is using the photoscreener as a refractive error measurement device. Also, the WA Spot photoscreener was mostly intended for pediatric use and your study sample is young adults.

We agree with the Reviewer that standard, cycloplegic refractive error measurement with the autorefractometer is better option than usage of photoscreener in an individual refractive error measuring. Although, we would like to point out the factor that have influenced our decision is that our study was planned at the height of COVID-19 pandemic and carried out still during the pandemic Usage of photoscreener that allows to scan refractive error from a longer distance than the autorefractometer was a safer option to screen the group consisting of hundreds of subjects that could be potentially infected. What is more, our study shows that performing a myopia screening is possible even during the events such as COVID-19 pandemic. Additionally, other studies confirm the high specifity and sensitivity of WA Spot vision screener (10.3928/01913913-20200331-02, 10.1016/J.JAAPOS.2014.07.176, 10.1016/J.JAAPOS.2018.09.012, 10.3390/IJERPH19148655/S1)- the sensitivity and specifity should be sufficient to describe the population’s general refractive status. To inform potential readers about this, we have added new information in the ,,Limitations of our study’’ section [lines 273-282].

Please, see the attachment- we are providing the version of the manuscript with the changes proposed by the Reviewer coloured in red.
